# Perceptions of social media challenges and benefits during the Covid-19 pandemic: Qualitative findings from a cross sectional international survey

Mariyana Schoultz[1], Gary Lamph[2], Hilde Thygesen[3,4]*, Janni Leung[5], Tore Bonsaksen[6,7], Mary Ruffolo[8], Daicia Price[8], Paul Watson[1], Isaac Kabelenga[9], Vivian Chiu[5], Amy Østertun Geirdal[10]

1 School of Health and Life Sciences, Northumbria University, Newcastle upon Tyne, United Kingdom, 2 School of Nursing, The University of Central Lancashire, Preston, United Kingdom, 3 Faculty of Health Sciences, Oslo Metropolitan University, Oslo, Norway, 4 Faculty of Health Studies, VID Specialized University, Oslo, Norway, 5 Faculty of Health and Behavioural Sciences, The University of Queensland, Brisbane, Australia, 6 Faculty of Social and Health Sciences, Department of Health and Nursing Science, Inland Norway University of Applied Sciences, Elverum, Norway, 7 Faculty of Health Studies, Department of Health, VID Specialized University, Stavanger, Norway, 8 School of Social Work, University of Michigan, Ann Arbor, MI, United States of America, 9 Department of Social Work and Sociology, School of Humanities and Social Sciences, University of Zambia (UNZA), Lusaka, Zambia, 10 Faculty of Social Sciences, Department of Social Work, Oslo Metropolitan University, Child Welfare and Social Policy, Oslo, Norway

* hilthy@oslomet.no

**Data Availability Statement:** A minimal anonymized data set is within the Supporting Information file.

## Abstract

Since the beginning of the Covid-19 pandemic in January 2020 the need for rapid information spread grew and social media became the ultimate platform for information exchange as well as a tool for connection and entertainment. With the rapid information spread along came the various public misconceptions and misinformation which consequently influenced perceptions and behaviors of the public towards the coronavirus pandemic. Thus, there was a need for identification and collation of public perceptions information to address future public health initiatives. This cross-national study aimed to examine the challenges and benefits of using social media during the Covid-19 pandemic outbreak. This study was a content analysis of the open-ended questions from a wider cross-sectional online survey conducted in Norway, UK, USA, and Australia during October/November 2020. 2368 participants out of 3474 respondents to the survey provided the open text responses included in the qualitative analysis. Thematic analysis was conducted independently by two researchers. All statements were coded to positive and negative sentiments. Three overarching themes were identified: 1. Mental health and emotional exhaustion 2. Information and misinformation; 3. Learning and inspiration.While providing a powerful mode of connection during the pandemic, social media also led to negative impact on public perceptions, including mistrust and confusion. Clarity in communications by institutions and education about credible information sources should be considered in the future. Further research is required in exploring and documenting social media narratives around COVID-19 in this and any subsequent incidents of pandemic restrictions. Understanding the public perceptions and their social narratives can support the designing of appropriate support and services for people in the future,

**Funding:** The authors received no specific funding for this work.

**Competing interests:** The authors have declared that no competing interests exist.

**Abbreviations:** CDC, Centre for Disease Control and Prevention; UK, United Kingdom; USA, United states of America; SARS, Severe Acute Respiratory Syndrome; WHO, World Health Organization.

while acknowledging the uncertainty and overwhelming impact of the pandemic that may have skewed the experiences of social media.

## Introduction

The first case related to the corona virus pandemic was identified in Wuhan, China in December 2019 [1]. Since, the world has seen dramatic changes to normal living in most countries. Common changes that have taken place include restricting social gatherings, local and international travelling, maintaining social distancing, and working from home just to mention a few [1,2]. This has led to devastating disruptions to the social and economic functioning of nations that is comparable to the impact of the Spanish flu pandemic of 1918 [3]. Since the initial outbreak, many of the restrictions for travel and social distancing remained (to various degrees) in most countries across Europe, America, and Australia with only a periodical easing off in some areas.

These long-term restrictions led to people having to spend more time at home with very little social interactions outside of their immediate households. For many, this meant increased job uncertainty and fear for their future livelihood [4], while others became less trusting of the governments and their management of the pandemic [5]. High Social Capital is a key determinant of positive health. Considering that social interactions and trust foster social capital, pandemic restrictions reduced that. Restrictions also negatively impacted mental health and contributed towards pandemic-related anxiety and depression [6]. All these events influenced how people socialized, worked, and interacted with each other as well as on how people coped and were able to access information. Hence, social media became even more so than pre-pandemic, the fastest and most accessible means for gathering and broadcasting information. It was used as a way of staying connected to family, friends and work and as a way of coping with life during the pandemic [6].

Social media provides many benefits and provides a source of information, entertainment and a source of connection via a cumulation of different electronic platforms [6]. Such messaging platforms include WhatsApp, SnapChat, Facebook, Twitter, Instagram, LinkedIn, TikTok, and YouTube. All of these outlets have billions of users worldwide and have had a major global impact on how people perceive life during the pandemic [6]. While many users of these platform are personal users, there has been a rapid organizational utilization of the platforms to support and disseminate reliable information. These organizations include the World Health Organization (WHO) and Centers of Disease Control and Prevention (CDC) as well as the local governments and health authorities in each country, each of which has gained hundreds of thousands of followers during the pandemic owing to the increased interest in decision making of local, national and international science and government policy directives aligned to the pandemic and restrictions.

As a result of the ongoing severity of the COVID-19 pandemic, there was a need for rapid information spread. However, the rapid information spread brought a new type of purposefully spread of disinformation and misinformation, particularly on social media platforms [7]. This infodemic contributed to public mistrust, fear, and misconceptions of the pandemic which consequently influenced the public perceptions and behaviors sometimes with a detrimental effect [8]. Thus, in order to dispel further misinformation and ensure the correct type of future public initiatives are designed, brough on an important need for identification and collation of public perceptions information. Hence, the aim of this study was to examine the public perceptions of the benefits and challenges of using social media during the Covid-19 pandemic outbreak.

## Methods

The consolidated criteria for reporting qualitative research (COREQ) were used to enhance the rigour and guide the structure of this paper [9].

### Ethics statement

The introductory information accompanying the survey explained the purpose and anonymity of the study and consent was obtained by completing the first page of the survey. The study was approved by OsloMet (20/03676) and the regional committees for medical and health research ethics (REK; ref. 132066) in Norway, reviewed by the University of Michigan Institutional Review Board for Health Sciences and Behavioral Sciences (IRB HSBS) and designated as exempt (HUM00180296) in USA, by Northumbria University Health Research Ethics (HSR1920-080) in UK and University of Queensland (HSR1920-080 2020000956) in Australia.

### Study design

This study was a content analysis of the open-ended questions from a wider on-line survey [10] about mental health, wellbeing, worry and the use of social media during the pandemic conducted in Norway, UK, USA, and Australia. The cross-sectional survey was available between October and November 2020. Invitations to take part in the survey were placed on social media such as Twitter, Facebook and Instagram in Norway, USA, UK, and Australia. Data was collected simultaneously in each country for approximately 4 weeks. Each country had a landing site for the survey at the following universities: OsloMet—Oslo Metropolitan University, Norway; University of Michigan, USA; Northumbria University, UK; and the University of Queensland, Australia.

### Participants and setting

AØG from OsloMet initiated the project, however each participating university/country had a project lead adhering to local ethical approvals. Participants in the study were from the general population in the participating countries. To participate in the survey, the participants had to: 1) be of age of 18 or over and 2) speak Norwegian or English.

### Questionnaire

The survey was simultaneously co-developed by the researchers in two languages, Norwegian and English, and was based on previous research conducted by the research group in the early phase (April 2020) of the Covid-19 pandemic [10,11].

The secure Qualtrics survey platform was used to design and disseminate the survey [12] which had 85 questions in total and took approximately 10 min to complete. Only the responses from the last two open-ended question are presented in this paper (see research question below). All open responses were reviewed in detail to identify common themes.

### Survey questions

1. Overall, what benefits have you experienced in using social media during the COVID-19 pandemic?

2. Overall, what challenges have you experienced in using social media during the COVID-19 pandemic?

## Analysis

Two researchers (MS, PW) independently coded the data to minimize subjectivity. All data were analysed using a thematic analysis approach [13]. Thematic analysis is a rigorous method of qualitative data analysis, consisting of six phases. These phases provide structure for the data to be organised, coded and a framework for theme identification.

Phase 1—Familiarizing yourself with your data

Data was inputted into an excel spreadsheet. The two researchers immersed themselves into this data by reading through it all twice for familiarization purposes.

Phases 2—Generating initial codes, and 3 -Searching for themes

From this data the two researchers independently generated initial codes and then searched for themes worthy of reporting. At this stage the researchers then met to discuss their findings and to extract the main themes for reporting.

Phase 4 –Reviewing themes, and Phase 5 –Defining and naming themes

Both researchers then reviewed the themes against the raw data and through discussion the themes were named to inform a wider research team report.

Phase 6 –Producing the report

A report on the findings was then presented to the wider research team and discussed.

## Rigour

Qualitative studies are often judged by trustworthiness and rigour, meaning that integrity and competence are required to be demonstrated within a study [14]. Analysis was conducted by two researchers independently which satisfies the above criteria. MS and PW met at three points (after the third, fifth and sixth step) to discuss findings and settle any potential disputes–there were none in this case. Rigour with sampling was ensured through maximum variation sampling (diverse sample) to make the data "information rich" [15].

## Results

### Demographics

A total of 2368 participants provided answer to the open-ended questions included for qualitative analysis. Table 1 includes the demographic data of the sample.

### Themes emerged

Boundaries between the themes reported are fluent rather than fixed, hence materials reported in one theme may also have relevance to others. All statements were coded to positive and negative sentiments. Three main themes were identified: 1. Mental health and emotional exhaustion 2. Information and misinformation; 3. Learning and inspiration.

**1. Mental health and emotional exhaustion.** This theme was the most prominent and commonly reported theme and had links to all the other themes. The general sentiment of this theme was the negative impact on mental health and the overall emotional exhaustion that people felt due to various reasons such as: negativity and internet addiction, stress, and anxiety. Each of these subthemes will be elaborated on and aligned with context to the data collected.

*Negativity and internet addiction.* Mainly, people felt that there was too much negativity on social media that made them feel overwhelmed which contributed to anxiety, distress, sadness, and had an overall negative impact on their mental health. Examples of which included:

> *'I get overwhelmed when negativity is rampant. Anxiety increases and, to a degree, sadness'. (Participant 684)*

**Table 1. Participant characteristics (N = 2368).**

| | | Total | Norway | UK | USA | Australia |
|---|---|---|---|---|---|---|
| | | N = 2368 | N = 542 | N = 364 | N = 1360 | N = 102 |
| | | % | % | % | % | % |
| Gender | | | | | | |
| | Male | 22.6 | 21.6 | 19.8 | 23.8 | 22.6 |
| | Female | 75.3 | 78.4 | 78.9 | 73.2 | 74.5 |
| | Other/prefer not to say | 2.1 | 0.0 | 1.4 | 3.1 | 2.9 |
| Age | | | | | | |
| | < 24 years | 8.6 | 7.5 | 14.4 | 7.7 | 5.1 |
| | 25–34 years | 23.1 | 17.0 | 19.7 | 27.5 | 10.1 |
| | 35–44 years | 19.0 | 21.6 | 21.7 | 17.5 | 16.2 |
| | 45–54 years | 16.2 | 19.8 | 19.4 | 13.6 | 20.2 |
| | 55–64 years | 15.1 | 14.8 | 18.1 | 13.7 | 24.2 |
| | 65–74 years | 14.1 | 13.6 | 4.7 | 16.2 | 23.2 |
| | >75 years | 3.8 | 5.7 | 1.9 | 3.7 | 1.0 |
| Education | | | | | | |
| | High school/technical degree | 26.99 | 25.64 | 32.78 | 25.96 | 27.45 |
| | Bachelor's degree | 35.64 | 35.53 | 34.71 | 36.03 | 34.31 |
| | > Master or above | 37.37 | 38.83 | 32.51 | 38.01 | 38.24 |
| Living area | | | | | | |
| | Rural or farming area | 14.2 | 7.7 | 18.8 | 16.7 | 0.0 |
| | In a town/ suburb | 46.7 | 36.2 | 35.7 | 56.2 | 15.8 |
| | In a city | 39.0 | 56.2 | 45.4 | 27.1 | 84.2 |

*'Too much . . .. negativity (and arguments on social media) having an overall negative impact on my mental health.' (Participant 154)*

For some people the negative interactions over social media had a more profound effect, in that it forced them to either block or unfriend people to protect their mental health.

*'Too much negative. People who I thought were my friends were posting negative comments on my feed, lots of hateful and racial bigotry. Had to unfriend people who were very close to me.' (Participant 364)*

Additionally, many participants reported and recognised the addictive nature of using social media use:

*'My experience is almost an addiction and a need to check for updates more than necessary.' (Participant 314)*

The addictive nature of social media check in was even compared to a smoking addiction, as underscored in the below quote:

*'It's become an addiction. First thing I do in the morning is check FB, and it's the last thing I do before bed. Similar to a smoker needing cigarette immediately after waking up.' (Participant 1433)*

Whilst the addictive reports of social media were common some participant also described how this also impacted negatively upon mental health and wellbeing, and that it influenced general outlook of life and the future.

*'Reading the comments on posts can be very stressful but oddly addictive, not good for mental health.' (Participant 146)*

*Stress and anxiety*. The general feeling is that many people found using social media more stressful than helpful. This was promoted by the different quarrels and disputes participant were reporting to have encountered on social media or the headlines that some media outlets were using:

*'I think social media during the pandemic has been more harmful than good. It usually makes me feel more stressed out than I was before I went on the social media.' (Participant 1378)*

*'Overall, it greatly increases stress. It's just a bunch of people yelling out into the echo chamber and compounding each other's anxieties. On the balance, I think it's very bad for my mental health.' (Participant 676)*

Others started to question if this heightened stress was due to quarantine or other factors rather than directly coming from their social media experience, hence questioning the catalyst for the social media negativity, stress, and anxieties. *The following quote underscore this*:

*'Trying to decide if what a person said genuinely made me this angry or if it's just quarantine stress.' (Participant 1020)*

Using social media brought for some feelings of fear, being scared and descriptions of this went as far as claiming it had a paralysing impact on allowing people to get on with their lives. For some, this fear was due to the constant influx of information about the dangers of Covid-19. For others, it was due to seeing people not following the guidelines:

*'I feel scared a lot when I see people not following guidelines.' (Participant 1058)*

The general feeling and mental health impact reported in this theme highlights how social media had contributed to the reported tiredness and emotional exhaustion, resulting from being overly connected with social media information and the associated heightened emotions reported by the participants. Additionally, navigating through the many harmful interactions that took place on social media further contributed to the emotional exhaustion reported. Also, people were fatigued of interaction via screens and the lack of face-to-face interactions.

*'tired of staring at screens.' (Participant 118)*

Participants report of making comparisons of themselves with the people/friends they follow on social media also contributed to stress and anxiety amongst our sample.

*'Emotional exhaustion from being overly connected to the stress and worry from the media, stress from watching friends and strangers become confident behind their keyboard leading to arguments, name calling, and other negative interactions . . . .' (Participant 463)*

**2. Information and misinformation.** This theme focused on information and misinformation, that was influenced by insights into participants views on control over information and its spread and the associated frustrations then encountered relating to misinformation.

*Control over information and its spread.* Many participants reported that they used social media to connect to friends and family or find news about their local area and news from around the world. Some said that they felt social media gave them control over information and how its spread.

> *'Information, e.g., from the newspapers are shared very fast, and it makes it easier to keep updated.' (Participant 2)*

> *'I felt good curating information for my friends and family.' (Participant 1025)*

On the other hand, others felt complete anger and frustration from the spread of misinformation, and even sadness about how information and misinformation caused division among people and somewhat reduced empathy and humanity. Many reported that while information-seeking online, the sheer volume of Covid-19 related information and misinformation, left them feeling stressed, worried, and overwhelmed.

> *'I find there is even more a blame culture, little empathy and miscommunication between people and disinformation.' (Participant* 340)

*Anger and frustration.* This sub-theme describes feelings of frustration and anger directed at political and mainstream media. Mistrust in relation to political issues is described locally, nationally and censorship of information were all felt to be causing division in societies views on the pandemic:

> *'The censorship of conservatives, and anyone who has a different opinion other than the current media narratives is disgusting and outrageous.' (Participant 342)*

Expressions of frustration and anger was shared amongst participants aligned to frustration due to political issues and of their countries handling of the pandemic:

> *'I have felt very frustrated and angry due to politics of people. I feel even more frustrated and angry with the families I see acting reckless and allowing their kids to hang out in large groups, not distancing and not masking, having meals in restaurants, sleepovers, etc and the rest of us have to isolate because of their selfish behaviour.' (Participant 377)*

Many reported additional frustrations owing to a portion of the population not taking the pandemic seriously and therefore not complying with national guidance. Mistrust in political and mainstream narratives from authorities and the media were felt to have individual and wider impacts, with some participants noting how political disagreements had negatively impacted upon friendships with some participants being compelled to unfriend people they would normally interact with. Some described how they had '*watched friendships end because of politics.' (Participant 1987).*

Others felt that there is too much censorship on Facebook and other social media platforms which made them feel like their rights have been infringed:

> *'Right or left is irrelevant. . . . but I hate the one sidedness and censorship that has developed over the past couple months.' (Participant 1834)*

A strong feeling is shared owing to participant-observed perceptions of censorship, which led to reservations in trusting their governments. Others felt deeply saddened by the division caused due to politics and bias.

*'Online bullying, especially for my political views. Censorship is also a problem on social media.' (Participant 1214)*

**3. Learning and inspiration.** Despite a majority of participants expressing strong negative emotions and experiences with social media during the pandemic, others emphasized its positive impacts. Through our analysis we identified four related sub-themes. Social media was by some experienced as a providing connection. Others described it as providing as a welcome distraction or providing inspiration by support new knowledge and skills acquisition, for others it was reported to provide self-help and self-care support.

*Providing connection*

*'It's been nice to keep in touch with my friends from college who live far away. There is a sense of comradery and community in that we're all doing this together.' (Participant 1146)*

Participants identified a striving for connection and search for a sense of community during the pandemic. However, conversely, others described the opposite and instead reported on how social media helped them feel closer to other people, helped their mood, reduced loneliness and, importantly, provided opportunities to be part of communities that they otherwise could have not encountered, due to accessibility and geography.

*'It's been so easy to feel disconnected from community and to feel lonely. To have a flourishing online group of people like me did wonders for my mood and my feelings of disconnection.' (Participant 1464)*

*A welcome distraction*. For some social media provided a welcome distraction from the daily stress of the pandemic and created for some new opportunities for development of new interests or engagement with past interests:

*'I have been inspired to create by the artists I follow on social media. Social media can also provide a welcome distraction from the stress I'm dealing with due to my college classes, my job, extracurriculars, and having to make important decisions regarding my future.' (Participant 897)*

*New knowledge and skill acquisition*. Other participants saw social media as providing a unique opportunity to enhance their knowledge about the world, nutrition, health or even business. For some of the participants this resulted in acquiring new hobbies or getting reacquainted with past interests and hobbies:

*'But I look at it this way—they say learning all these new things is good for my aging brain. Hoping Facebook doesn't disappear, but if it does, I'll probably learn to use some other social media.' (Participant 679)*

*Self-help and self-care*. For some participants social media has been a helpful tool in learning more about self-care and self-help activities, while for others it has provided inspiration for home renovation and DIY projects. While challenging to adapt to at the start of the pandemic,

social media has, for some, turned out to be a great way of learning not only new digital skills for connecting, but also in guiding and supporting new physical activities that has helped them stay active.

*'Keeping new thoughts and ideas fresh on a daily basis. 'Learn something new each day. Entertainment is delightful.' (Participant 193)*

## Discussion

The primary aim of this study was to explore the public perceptions on benefits and challenges of using social media during the timeframe when lockdown and restrictions were most severe in the participating countries.

The findings present a strong view and unique insight relating to the negative impact the pandemic had upon people's mental health and wellbeing, which has been compounded by feelings of negativity, fear, stress, anxiety, and emotional tiredness. However, the findings also reveal that for some people, the use of social media has been an important resource for learning and inspiration.

Social media usage has increased since the start of the pandemic [16]. Whilst it has been established to be a useful mode for information sharing and updates, this has been conflicted by the spread of misinformation and 'fake news', leading to extra anxiety and public confusion [17,18]. These polarised views of social media use and perceptions are evident in our research. Due to the wealth and 'barrage' of information on social media, the public was sometimes led to ill-informed decision-making processes. For example, the sheer volume of information and conflicting messages was challenging for individuals having to interpret and understand this information in order to form their own views, opinions and to inform their responses to the first pandemic to occur in a century [10,18]. The high and frequent use of social media over the course of the pandemic has been identified as having a negative impact on mental health, wellbeing, and loneliness [19,20]. Whilst social media allowed news to be spread fast, it was also compromised by the lack of governance and misinformation shared based on personal opinions. Thus, Social Media Guidance and signposting to credible sources of information would have been useful for social media users. Additionally, educational institutions could have responded to this by providing informative support in spotting credible sources of information and differentiating from 'fake news' [20].

The rapidly changing landscape of COVID-19 information sharing could be partially responsible for opening the door to negative misinformation from non-credible sources. Nonetheless, information and misinformation, has undoubtably contributed to the feeling of stress, anxiety and participants reports of feeling emotionally overwhelmed.

Politics and lack of trust in governments and institutions were often reported in the qualitative data, which feed into this confusion [20]. The findings of this study provide insight into the key challenges around social media use and their impact upon mental health during and after the initial pandemic lockdowns. These concerns were further compromised via fear that came from social media, and the persistent uncertainty/changing status of the pandemic and associated guidance. Subsequently this led to concerns relating to income security, thus leading to some parts of society choosing to ignore guidance and breaking of rules, owing to real societal fears such as job losses [21].

Our results also need to be considered in the context of the individual participants. Those expressing frustration by social media may have had confidence in the guidance provided, and those who experienced minimal disruptions to their roles, income or employment are likely to report the responses differently to those who feel unsupported or let down by the authorities.

It has however been found that those who were remote working reported better overall mental health outcomes owing to the ability to still carry out occupational duties outside of the workplace [21]. Hence the experiences and judgements shared are likely to be influenced by participants own unique social circumstances. However, despite this, our quantitative data provided evidence of a split in trust of local and national handlings of the pandemic, with higher levels of trust reported in Norway and Australia, and much lower levels of trust reported in the UK and USA [20].

At the same time, it is important to emphasize that social media has also played an important role in providing a platform for connecting people during the lockdown, that without this would have left people feeling even more isolated. Passive social media engagement e.g., aimlessly scrolling and information sourcing can result in increased loneliness and lowered life satisfactions [22,23]. Hence increased social media engagement during the pandemic could have inadvertently resulted in negative impacts and consequences for some of our participants.

In our previous study [24] we mainly focused on the negative emotional impact of the pandemic. However, as the pandemic prolonged into subsequent lockdowns, the views of participants have expanded into individual perspectives of the government handlings, responses and political frustrations that have arose. Growing social frustrations, debates and polarised personal opinions relating to the pandemic have emerged amongst the cross-national sample and there was clear evidence of frustration and in some cases highlighted stress and distress.

The pandemic appears to have created opportunities for some of our participants who reported that they gained new digital skills and ability to engage with social media forums. However, it should be noted that many older people had reservations of engaging with social media and digital platforms owing to the cultural differences, such as online application of social etiquette and the technological skills and knowledge to effectively engage [24].

## Strengths and limitation

A strength of this study is the diverse sample of participants in the four countries and the method of qualitative thematic analysis we have undertaken that has explored the benefits and challenges associated with using social media by the multinational population since the start of the pandemic. The findings from this study expand upon the findings of this research teams first survey at the start of the pandemic [10,24], and whilst similar themes have been identified, the complexity of qualitative reporting has been increased and also displays the emerging shift in attitudes towards the handling of the pandemic.

This study is not without its limitations. Selection bias may have been encountered as those who participated were self-selected. Hence it could be argued that only those who have the strongest feelings about the pandemic took time to complete this survey. This survey was completed digitally. Therefore, we are likely to have received only the opinions of those with access to and those most comfortable with technology completing the surveys. Only two open questions that were focused on social media experiences during the pandemic were included in the survey, and hence drawing conclusions from this data is unlikely to be representative of whole populations. Future research should consider in-depth interviews or focus groups to further explore why some people felt more connected with social media while some others had increased anxiety. But despite this, the data does provide some useful insights not previously explored in a cross-national format.

## Conclusion

While providing a powerful mode of connection during the pandemic, social media use also led to negative impact on public perceptions, including mistrust and confusion. Clarity in

communications by institutions and education about credible information sources should be considered in the future. Further research is required in exploring and documenting the social media narratives around COVID-19 in western societies in this and any subsequent incidents of pandemic restrictions. Understanding the public perceptions and their social narratives can support the designing of appropriate support and services for people in the future, while acknowledging the uncertainty and overwhelming impact of the pandemic that may have skewed the experiences of social media. When harnessing technology for benefits, further considerations should be considered to minimize the adverse impact of using social media on users.

## Supporting information

**S1 Data. Minimal anonymized dataset.**
(SAV)

## Acknowledgments

The authors acknowledge the time and efforts spent by the participants when responding to the survey.

## Author Contributions

**Conceptualization:** Mariyana Schoultz, Gary Lamph, Hilde Thygesen, Janni Leung, Tore Bonsaksen, Mary Ruffolo, Daicia Price, Paul Watson, Vivian Chiu, Amy Østertun Geirdal.

**Formal analysis:** Mariyana Schoultz, Hilde Thygesen, Janni Leung, Paul Watson, Vivian Chiu.

**Investigation:** Mariyana Schoultz, Hilde Thygesen, Janni Leung, Tore Bonsaksen, Mary Ruffolo, Daicia Price, Paul Watson, Vivian Chiu, Amy Østertun Geirdal.

**Methodology:** Mariyana Schoultz, Gary Lamph, Hilde Thygesen, Janni Leung, Tore Bonsaksen, Mary Ruffolo, Daicia Price, Paul Watson, Vivian Chiu, Amy Østertun Geirdal.

**Project administration:** Amy Østertun Geirdal.

**Supervision:** Amy Østertun Geirdal.

**Validation:** Mariyana Schoultz, Gary Lamph, Hilde Thygesen, Janni Leung, Tore Bonsaksen, Mary Ruffolo, Daicia Price, Isaac Kabelenga.

**Visualization:** Amy Østertun Geirdal.

**Writing – original draft:** Mariyana Schoultz, Gary Lamph, Hilde Thygesen, Janni Leung, Paul Watson, Vivian Chiu.

**Writing – review & editing:** Mariyana Schoultz, Gary Lamph, Hilde Thygesen, Janni Leung, Tore Bonsaksen, Mary Ruffolo, Daicia Price, Isaac Kabelenga, Amy Østertun Geirdal.

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
