## [Decision Letter · Decision Letter 0]

14 Sep 2022

PGPH-D-22-00563

Perceptions of social media challenges and benefits during the Covid-19 pandemic: qualitative findings from an international survey

Dear Dr. Thygesen,

Thank you for submitting your manuscript to PLOS Global Public Health. After careful consideration, we feel that it has merit but does not fully meet PLOS Global Public Health’s publication criteria as it currently stands. Therefore, we invite you to submit a revised version of the manuscript that addresses the points raised during the review process.

The paper reads well. However, it is unclear why you refer your study design as a mixed - method. It seems to me this was a survey whereby you had a mix of closed and open-ended questions.  You conducted content analysis of the open-ended questions.  That should be clarified.

We look forward to receiving your revised manuscript.

Kind regards,

Joel Msafiri Francis, MD, MS, PhD

Academic Editor

Journal Requirements:

1. Please send a completed 'Competing Interests' statement, including any COIs declared by your co-authors. If you have no competing interests to declare, please state "The authors have declared that no competing interests exist". Otherwise please declare all competing interests beginning with the statement "I have read the journal's policy and the authors of this manuscript have the following competing interests:"

i. Please clarify all sources of funding (financial or material support) for your study. List the grants (with grant number) or organizations (with url) that supported your study, including funding received from your institution. 

ii. State the initials, alongside each funding source, of each author to receive each grant.

Additional Editor Comments (if provided):

The paper reads well. However, it is unclear why you refer your study design as a mixed - method. It seems to me this was a survey whereby you had a mix of closed and open-ended questions. You conducted content analysis of the open-ended questions. That should be clarified.

Reviewers' comments:

Reviewer's Responses to Questions

**Comments to the Author**

1. Does this manuscript meet PLOS Global Public Health’s publication criteria? Is the manuscript technically sound, and do the data support the conclusions? The manuscript must describe methodologically and ethically rigorous research with conclusions that are appropriately drawn based on the data presented.

Reviewer #1: Yes

Reviewer #2: Yes

2. Has the statistical analysis been performed appropriately and rigorously?

Reviewer #1: N/A

Reviewer #2: Yes

3. Have the authors made all data underlying the findings in their manuscript fully available (please refer to the Data Availability Statement at the start of the manuscript PDF file)?

Reviewer #1: No

Reviewer #2: Yes

4. Is the manuscript presented in an intelligible fashion and written in standard English?

Reviewer #1: Yes

Reviewer #2: Yes

5. Review Comments to the Author

Reviewer #1: This is a qualitative study to understand the effect and role of social media during the COVID-19 pandemic. The topic is important due to increased internet and social media use during the pandemic. Here is my comments:

1/ Qualitative studies do not in general aim to generalize the results. The authors may want to justify the data collection methods. In fact, in depth interview to some of this participants may explain much further more. E.g. why some people felt more connected with social media while some others had increased anxiety. I suppose because it is online questionnaire, the answers will often be short and it is often not possible fully understand an individual story. I think individual interviews and focus groups may achieve this aim much better

2/ As the authors also mentioned that this is part of a mixed method study, how are the results "mixed"? at what level? how do the quantitatively and qualitative results interact? It is ok not to include all the results in one study (often due to limited word count) but the authors can show where we can find the mixed results

3/ Along the same thought, is triangulation possible? This may enhance the credibility of your results.

Reviewer #2: I am glad to avail the opportunity to review this creative study entitled, “Perceptions of social media challenges and benefits during the Covid-19 pandemic: qualitative findings from an international survey.”

This study describes that since the beginning of the Covid-19 pandemic in January 2020 and the various national restrictions on social contact and travel in Europe, America and Australia,

people became dependent on social media as a means for gathering information and a tool for staying connected to family, friends, and work. However, few cross-national qualitative studies have been done on the challenges and benefits of use of social media during the first wave of Covid-19. This cross-national study aimed to qualitatively examine the challenges and benefits of using social media during the Covid-19 pandemic outbreak.

I want to accept this study for publication. However, the authors need to revise it according to my suggestions. The title needs clarity with the design of the main study. I suggest authors revise their titles with better and suitable words. See the below-recommended studies to improve your TTLE and Abstract quality.

Abstract

First, I have some suggestions for the authors to enhance the quality of this innovative study. Please write a high-quality abstract, as it is the main door of the study. I suggest authors add Graphical Abstract in a meaningful way to reflect the whole idea. The abstract should be in a structured format.

Introduction section

This section needs improvement. Please read these studies, revise your abstract, and cite them ithe introduction and literature part. Cite the suggested studies to improve the quality.

Su, Z., McDonnell, D., Wen, J., Kozak, M., Segalo, S., Li, X., Ahmad, J., Cheshmehzangi, A., Cai, Y., Yang, L., & Xiang, Y. T. (2021, Jan 5). Mental health consequences of COVID-19 media coverage: the need for effective crisis communication practices. Global Health, 17(1), 4. https://doi.org/10.1186/s12992-020-00654-4

Azadi, N. A., Ziapour, A., Lebni, J. Y., Irandoost, S. F., & Chaboksavar, F. (2021). The effect of education based on health belief model on promoting preventive behaviors of hypertensive disease in staff of the Iran University of Medical Sciences. Archives of Public Health, 79(1), 69. doi:10.1186/s13690-021-00594-4

Maqsood, A., Rehman, G., & Mubeen, R. (2021, 2021/11/01/). The paradigm shift for educational system continuance in the advent of COVID-19 pandemic: Mental health challenges and reflections. Current Research in Behavioral Sciences, 2, 100011. https://doi.org/https://doi.org/10.1016/j.crbeha.2020.100011

Abbas, J. (2020). The Impact of Coronavirus (SARS-CoV2) Epidemic on Individuals Mental Health: The Protective Measures of Pakistan in Managing and Sustaining Transmissible Disease. Psychiatr Danub, 32(3-4), 472-477. https://doi.org/10.24869/psyd.2020.472

Literature

In the emergence of the COVID-19 pandemic, healthcare systems have faced a tremendous pressure worldwide. This pandemic has affected all lifestyles. The study investigated an interesting research topic. The study offers interesting information and provides useful insight. I suggest authors cite latest literature related to COVID-19 pandemic effects to support the study. The suggested studies are;

Rahmat, T. E., Raza, S., Zahid, H., Mohd Sobri, F., & Sidiki, S. (2022). Nexus between integrating technology readiness 2.0 index and students’ e-library services adoption amid the COVID-19 challenges: Implications based on the theory of planned behavior. J Educ Health Promot, 11(1), 50. doi:10.4103/jehp.jehp_508_21

NeJhaddadgar, N., Ziapour, A., Zakkipour, G., Abolfathi, M., & Shabani, M. (2020, Nov 13). Effectiveness of telephone-based screening and triage during COVID-19 outbreak in the promoted primary healthcare system: a case study in Ardabil province, Iran. Z Gesundh Wiss, 1-6. https://doi.org/10.1007/s10389-020-01407-8

Yoosefi Lebni, J., Moradi, F., Salahshoor, M. R., Chaboksavar, F., Irandoost, S. F., Nezhaddadgar, N., & Ziapour, A. (2020, Jul 2). How the COVID-19 pandemic effected economic, social, political, and cultural factors: A lesson from Iran. Int J Soc Psychiatry, 20764020939984. https://doi.org/10.1177/0020764020939984

Su, Z., McDonnell, D., Cheshmehzangi, A., Li, X., & Cai, Y. (2021). The promise and perils of Unit 731 data to advance COVID-19 research. BMJ Global Health, 6(4).

Methods and Results

Please read the suggested studies and improve your method design and result sections. Cite these studies in methods and results to support the literature.

Li, Zhenhuan , Dake Wang, Kaifeng Duan, and R. Mubeen. 2021. "Social media efficacy in crisis management: Effectiveness of non-pharmaceutical interventions to manage the COVID-19 challenges." Front Psychiatry 12 (1099):626134. doi: 10.3389/fpsyt.2021.626134.

Aman, J., Nurunnabi, M., & Bano, S. (2019). The Impact of Social Media on Learning Behavior for Sustainable Education: Evidence of Students from Selected Universities in Pakistan. Sustainability, 11(6), 1683. http://www.mdpi.com/2071-1050/11/6/1683

Aqeel, M., Rehna, T., Shuja, K. H. (2022). Comparison of Students' Mental Wellbeing, Anxiety, Depression, and Quality of Life During COVID-19's Full and Partial (Smart) Lockdowns: A Follow-Up Study at a 5-Month Interval. Front Psychiatry, 13, 835585. doi:10.3389/fpsyt.2022.835585

Farzadfar, F., Naghavi, M., Sepanlou, S. G., Saeedi Moghaddam, S., Dangel, W. J., Davis Weaver, N., . . . Larijani, B. (2022). Health system performance in Iran: a systematic analysis for the Global Burden of Disease Study 2019. The Lancet. doi:10.1016/s0140-6736(21)02751-3

Discussion

Discussion is explained well. Check grammar errors in this section. I suggest authors cite latest literature related to COVID-19 pandemic effects to support the study. The suggested studies are;

Aqeel, M., Raza, S., & Aman, J. (2021). Portraying the multifaceted interplay between sexual harassment, job stress, social support and employees turnover intension amid COVID-19: A Multilevel Moderating Model. Foundation University Journal of Business & Economics, 6(2), 1-17. doi:https://fui.edu.pk/fjs/index.php/fujbe/article/view/551

Abbas, J., Wang, D., Su, Z., & Ziapour, A. (2021). The Role of Social Media in the Advent of COVID-19 Pandemic: Crisis Management, Mental Health Challenges and Implications. Risk Manag Healthc Policy, 14, 1917-1932. doi:10.2147/RMHP.S284313

Geng, J., Ul Haq, S., Ye, H., Shahbaz, P., Abbas, A., & Cai, Y. (2022). Survival in Pandemic Times: Managing Energy Efficiency, Food Diversity, and Sustainable Practices of Nutrient Intake amid COVID-19 Crisis. Frontiers in Environmental Science, 13, 945774. doi:10.3389/fenvs.2022.945774

Yu, S., Draghici, A., Negulescu, O. H., & Ain, N. U. (2022). Social Media Application as a New Paradigm for Business Communication: The Role of COVID-19 Knowledge, Social Distancing, and Preventive Attitudes. Frontiers in Psychology, 13. doi:10.3389/fpsyg.2022.903082

Conclusion

The conclusion section needs improvement and authors need to expand it as it will improve the quality of this study. The English level needs some improvement to reach a satisfactory level, specifically the grammar. It should sufficiently meet quality to reach scientific merit for publication. I recommend that the authors describe the study's scientific contribution to the existing body of knowledge in the discussion section. How does this study’s implications provide useful information for the scientific readership? I endorse this manuscript for publication after minor corrections, as suggested.

6. PLOS authors have the option to publish the peer review history of their article (what does this mean?). If published, this will include your full peer review and any attached files.

**Do you want your identity to be public for this peer review?** For information about this choice, including consent withdrawal, please see our Privacy Policy.

Reviewer #1: No

Reviewer #2: No

---

## [Decision Letter · Decision Letter 1]

12 Dec 2022

Perceptions of social media challenges and benefits during the Covid-19 pandemic: qualitative findings from a cross sectional international survey.

PGPH-D-22-00563R1

Dear Professor Thygesen,

We are pleased to inform you that your manuscript 'Perceptions of social media challenges and benefits during the Covid-19 pandemic: qualitative findings from a cross sectional international survey.' has been provisionally accepted for publication in PLOS Global Public Health.

Best regards,

Joel Msafiri Francis, MD, MS, PhD

Academic Editor

Reviewer Comments (if any, and for reference):

Reviewer's Responses to Questions

**Comments to the Author**

1. If the authors have adequately addressed your comments raised in a previous round of review and you feel that this manuscript is now acceptable for publication, you may indicate that here to bypass the “Comments to the Author” section, enter your conflict of interest statement in the “Confidential to Editor” section, and submit your "Accept" recommendation.

Reviewer #1: All comments have been addressed

Reviewer #2: All comments have been addressed

2. Does this manuscript meet PLOS Global Public Health’s publication criteria? Is the manuscript technically sound, and do the data support the conclusions? The manuscript must describe methodologically and ethically rigorous research with conclusions that are appropriately drawn based on the data presented.

Reviewer #1: Yes

Reviewer #2: Yes

3. Has the statistical analysis been performed appropriately and rigorously?

Reviewer #1: N/A

Reviewer #2: Yes

4. Have the authors made all data underlying the findings in their manuscript fully available (please refer to the Data Availability Statement at the start of the manuscript PDF file)?

Reviewer #1: Yes

Reviewer #2: Yes

5. Is the manuscript presented in an intelligible fashion and written in standard English?

Reviewer #1: Yes

Reviewer #2: Yes

6. Review Comments to the Author

Reviewer #1: Thank you. I think my concerns are addressed. In particular, the authors no longer frame it as a mixed-method study.

Reviewer #2: I have reviewed the revised manuscript entitled, "Perceptions of social media challenges and benefits during the Covid-19 pandemic: qualitative findings from a cross sectional international survey."

I endorse it for publication in its current format. Good luck

7. PLOS authors have the option to publish the peer review history of their article (what does this mean?). If published, this will include your full peer review and any attached files.

**Do you want your identity to be public for this peer review?** For information about this choice, including consent withdrawal, please see our Privacy Policy.

Reviewer #1: No

Reviewer #2: No
